# Capnophilic Lactic Fermentation from *Thermotoga neapolitana*: A Resourceful Pathway to Obtain Almost Enantiopure L-lactic Acid

**Genoveffa Nuzzo** †, **Simone Landi** †, **Nunzia Esercizio, Emiliano Manzo, Angelo Fontana** * and **Giuliana d'Ippolito** *

Bio-Organic Chemistry Unit, Institute of Biomolecular Chemistry CNR, Via Campi Flegrei 34, Pozzuoli, 80078 Naples, Italy; nuzzo.genoveffa@icb.cnr.it (G.N.); s.landi@icb.cnr.it (S.L.); esercizionunzia@gmail.com (N.E.); emanzo@icb.cnr.it (E.M.)

* Correspondence: afontana@icb.cnr.it (A.F.); gdippolito@icb.cnr.it (G.d.); Tel.: +39-081-8675096 (A.F.); +39-081-8675075 (G.d.)

† These authors have equally contributed to this work.

**Abstract:** The industrial production of lactic acid (LA) is mainly based on bacterial fermentation. This process can result in enantiopure or racemic mixture according to the producing organism. Between the enantiomers, L-lactic acid shows superior market value. Recently, we reported a novel anaplerotic pathway called capnophilic lactic fermentation (CLF) that produces a high concentration of LA by fermentation of sugar in the anaerobic thermophilic bacterium *Thermotoga neapolitana*. The aim of this work was the identification of the enantiomeric characterization of the LA produced by *T. neapolitana* and identification of the lactate dehydrogenase in *T. neapolitana* (TnLDH) and related bacteria of the order Thermotogales. Chemical derivatization and GC/MS analysis were applied to define the stereochemistry of LA from *T. neapolitana*. A bioinformatics study on TnLDH was carried out for the characterization of the enzyme. Chemical analysis showed a 95.2% enantiomeric excess of L-LA produced by *T. neapolitana.* A phylogenetic approach clearly clustered the TnLDH together with the L-LDH from lactic acid bacteria. We report for the first time that *T. neapolitana* is able to produce almost enantiopure L-lactic acid. The result was confirmed by bioinformatics analysis on TnLDH, which is a member of the L-LDH sub-family.

**Keywords:** optical purity; lactate dehydrogenase; stereochemistry; fermentation; thermophilic bacteria

## 1. Introduction

Lactic acid (2-hydroxypropanoic acid) (LA) is a naturally occurring organic acid that can be produced by fermentation or chemical synthesis. It is widely used in food, pharmaceuticals, cosmetics, and industrial applications, especially as a building block of bioplastic, i.e., polylactic acid (PLA). In recent years, global demand for LA and PLA has significantly increased, thus making production an impelling challenge of great interest [1]. For example, LA industrial production ranged from 150,000 metric tons in 2007 to 259,000 metric tons in 2012, with a commercial value of 1.57 US dollars $Kg^{-1}$ for a product with 88% purity [2]. Lactic acid is an enantiomeric molecule, and its production can lead to different grades of product, from pure enantiomers to racemic mixtures. Among these compounds, enantio-enriched L-lactic acid has a significant superior market value, whereas little consideration is given to racemic preparations. Nowadays, there is no industrial approach to synthesize by chemical methods pure stereoisomers; thus, global manufacturing is mostly based on carbohydrate fermentation by homofermentative lactic acid bacteria (LAB) from genera *Lactobacillus* and *Lactococcus* [3]. Fermentation

in these organisms relies on sugar catabolism by the Embden–Meyerhof–Parnas pathway and conversion of pyruvic acid to lactic acid by two different types of lactate dehydrogenase (LDH) named L-LDH and D-LDH according to the chirality of the downstream stereoisomers released in the culture broth [4].

Recently, we reported an efficient production of lactic acid by a novel anaplerotic pathway called capnophilic lactic fermentation (CLF) in the anaerobic bacterium *Thermotoga neapolitana* [5,6]. This organism is a hyperthermophilic gram-negative marine member of the Thermotogales order that is well recognized for a very efficient production of $H_2$ [7,8]. Glucose conversion to lactic acid by CLF is stimulated by $CO_2$ sparging and occurs without affecting $H_2$ synthesis [9]. The overall result is a high yield of $H_2$ (as much as 3.6 mol) and lactic acid (up to 0.6 mol) from one mole of glucose. This unexpected result paves the way for suitable biotechnological possibilities for the simultaneous production of $H_2$ and lactic acid from sugar-enriched agro-food residues [10,11].

The aim of the present study was the stereochemical characterization of LA produced by *T. neapolitana*, as well as the bioinformatic analysis of the LDH in this organism and taxonomically related members of the Thermotogales order. The investigation has been carried out on *Thermotoga neapolitana* subsp. *capnolactica* (DSM 33003), an efficient CLF strain recently described by our group [12].

## 2. Materials and Methods

### 2.1. Chemicals

Sodium L-lactate (purity: $\geq$ 99%, enantiomeric ratio: $\geq$99:1), sodium DL-lactate (purity: $\geq$ 99%, enantiomeric ratio: 55:45) (1R,2S,5R)-(−)-menthol (purity: > 99%) ((−)-menthol) and Trifluoroacetic anhydride (purity: > 99%) and acetyl chloride were from Sigma Aldrich (Milan, Italy).

### 2.2. Culture Conditions

Anaerobic culture of *T. neapolitana* subsp. *capnolactica* (*T. neapolitana* BCU1801–DSM 33003) was routinely maintained on medium supplemented with 0.5% glucose (w/v) and 0.4% (w/v) yeast extract/tryptone [7]. Precultures (25 mL) were incubated overnight at 80 °C without shaking and used to inoculate (6% v/v) 3.8 L fermenters with working volume of 1 L and headspace of 2.8 L. Cultures were stirred (250 rpm) at 80°C and sparged for 5 min with pure $CO_2$ at the beginning of the growth. After gas sparging, pH was adjusted at 7.5 by 1 M NaOH. Cells were collected at 24 h, centrifuged at 16,000 g 15 min (Hermle Z3236K). Supernatant was kept to -20°C until analysis. Cell growth was determined by optical density (OD) at 540 nm (UV/Vis spectrophotometer DU 730, Beckman Coulter). Three biological replicates were used for chemical analysis.

### 2.3. Chemical Analysis

Measurement of $H_2$ was performed by gas chromatography (GC) on an instrument (Focus GC, Thermo Scientific) equipped with a thermoconductivity detector (TCD) and fitted with a 3 m molecular sieve column (Hayesep Q), using $N_2$ as gas carrier. The glucose concentration was determined by the dinitrosalicylic acid method calibrated on a standard solution of 1 $gL^{-1}$ glucose [7]. Organic acids were measured by ERETIC [1]H NMR applied on culture broth after dilution of supernatant (0.6 mL) with 0.1 mL of $D_2O$ and transfer to an NMR tube. All experiments were performed on Bruker DRX 600 spectrometer equipped with an inverse TCI CryoProbe. Peak integration, ERETIC measurements and spectrum calibration were obtained by the specific subroutines of Bruker Top-Spin 3.1 program. Spectra were acquired with the following parameters: flip angle 90°, repetition time 20 s, SW = 3000 Hz, SI = 16K, NS = 16, RG = 1. An exponential multiplication was applied to the FID, inducing a line broadening of 1 Hz. No baseline correction was used.

### 2.4. Derivatization of Lactic Acid to the O-Trifluoroacetylated-(−)-Menthyl Ester

For the stereochemical analysis of natural lactic acid, we followed the O-trifluoroacetyl-(–)-methylation method by Inoue et al. [13]. Briefly, 200 microliters of (−)-menthol solution (200μg/μL in ethyl acetate) was added to lactic acid (~30 mg). After removing the organic solvent under a gentle stream of nitrogen, 200 μL of toluene and 10 μL of acetyl chloride were added to the residue. The mixture was then heated to 100 °C for 1 h. After heating, the excess reagent was removed under nitrogen. The residue was then reacted with 200 μL trifluoroacetic anhydride in acetonitrile (2:1) at 60 °C for 20 min. The excess reagent was removed by evaporation under nitrogen and the residue was dissolved in 5 mL chloroform for GC/MS analysis.

### 2.5. GC/MS Analysis

An aliquot (1 μL) of derivatized sample was injected into a GC/MS instrument equipped with a fused silica capillary column (Agilent VF-5ms capillary column 30 × 0.25 mm, 0.25 μm). The temperatures of the GC injector and of the interface line were 250 and 280 °C, respectively. After three minutes of isocratic condition, the column oven temperature was programmed to increase from 60 to 320 °C at a rate of 17 °C/min. The mass spectrometer was operated in electron ionization (EI) mode with 70 eV of electron energy.

### 2.6. Bioinformatics

Sequences of lactate dehydrogenase (LDH) from different bacteria representing genera (*Lactobacillus*, *Streptococcus*, *Staphylococcus*, *Sulfolobus*, *Escherichia* and others) were found using keywords and BLAST searches on UniProt database. Lactate dehydrogenase of Thermotogales were obtained on the Ensamble bacteria database (https://bacteria.ensembl.org/index.html) (access on December 2018). using a BLASTp approach. Alignments and phylogenetic analyses were performed using the software MEGA version 6 [14]. Sequences alignment was achieved using the MUSCLE algorithm, using a maximum number of interaction equal to 32. Phylogenetic tree was constructed using the maximum likelihood method with the substitution LG+I model gamma distributed. The test of phylogeny was performed using the bootstrap method with the number of replication equal to 100.

## 3. Results

### 3.1. Fermentation of T. Neapolitana Under CLF Conditions

*T. neapolitana* subsp. *capnolactica* was grown in 3.8 L fermenter under $CO_2$ sparging. After 24 h, cells consumed 3.5 g/L of glucose (Table 1). The $CO_2$ sparging led to a 1.14 and 0.7 yield (mol/mol of consumed glucose) of acetic acid and lactic acid, respectively. Hydrogen yield was 2.5 mol/mol of consumed glucose. In accordance with Di Pasquale et al. [6], lactic acid production was obtained through the activation of the CLF pathway by the coupling of acetate and $CO_2$.

**Table 1.** Fermentation parameters of *Thermotoga neapolitana* cultures grown under capnophilic lactic fermentation (CLF) conditions in 3.8 L fermenter for 24 h.

| Growth Parameters | | | | Yields (mol/mol glucose) | | |
|---|---|---|---|---|---|---|
| OD 540 nm | Dry Biomass (g/L) | Glucose Consumption (g/L) | $H_2$ | Acetic Acid | Lactic Acid | AL/AA |
| 0.79 ± 0.06 | 0.290 ± 0.03 | 3.5 ± 0.2 | 2.5 ± 0.3 | 1.14 ± 0.05 | 0.7 ± 0.02 | 0.58 |

### 3.2. Stereochemistry Determination

In agreement with Inoue et al. [13], retention time of both enantiomers of lactic acid was established on a GC column by comparing O-trifluoroacetyl-(−)-menthyl esters of DL-lactic acid and pure standard of L-lactic acid. The diagnostic fragmentations (*m/z* 141, 139 and 95) of

LA were considered for the unambiguous identification of the peak of interest in the total ion chromatogram (TIC). Natural lactic acid was recovered from fermentation broth of *T. neapolitana* cultures maintained under CLF conditions and analyzed as described above after derivatization in O-trifluoroacetyl-(−)-menthyl ester (Figure 1). The results clearly indicated the L-configuration of the natural product. Co-elution of DL-lactate and natural lactate showed an increase in the area relative to the peak of L-lactate derivative, thus confirming the assignment (Figure S1, Supplementary Information).

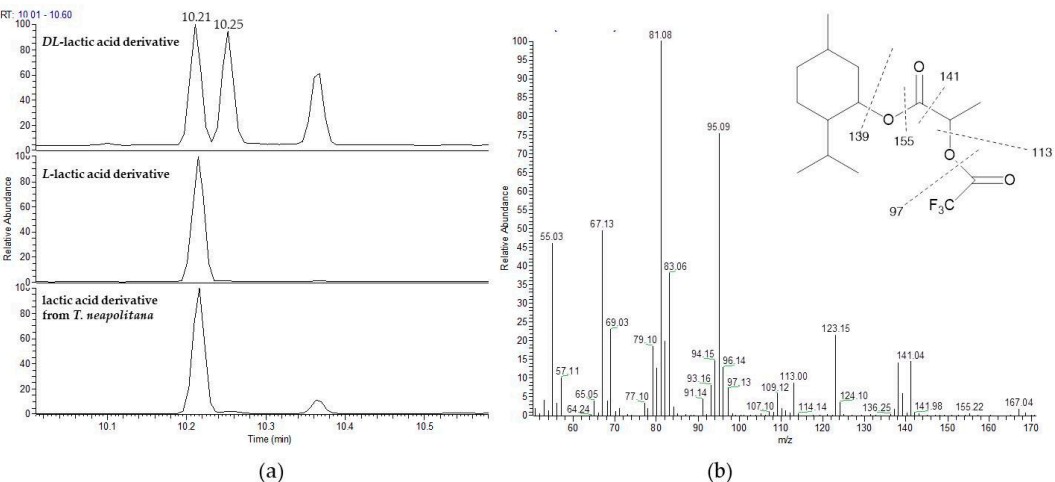

(a)                                    (b)

**Figure 1.** (**a**) The GC/MS of O-trifluoroacetyl-(–)-menthyl derivatives of DL-Lactic acid, L-lactic acid and natural lactic acid from *T. neapolitana*. Chromatograms show L-lactic and D-lactic acid derivatives at Rt of 10.21 min and 10.25 min, respectively; (**b**) mass spectrum and diagnostic fragmentations of the O-trifluoroacetyl-(−)-menthyl ester of lactic acid.

A small percentage of D-lactic acid was also found in the natural sample (Figure S2, Table 2); however, relative peak area of L- and D- derivatives in the natural metabolite established more than 95% of enantiomeric excess (e.e.) of the L- stereoisomer (Table 2).

**Table 2.** Optical purity of L-lactic acid from *T. neapolitana* cultures under CLF conditions. Peak area of L- and D- derivatives were obtained by GC/MS. The experiment was carried out in duplicate. e.e.: enantiomeric excess.

|           | Peak Area    | Relative AREA % | % e.e. |
|-----------|--------------|-----------------|--------|
| L-Lactate | 44618648.45  | 97.6            | 95.2   |
| D-Lactate | 1088407.05   | 2.4             | -      |

*3.3. Bioinformatic Approach to Characterize the Lactate Dehydrogenase of T. Neapolitana (TnLDH)*

The stereochemical assignment of LA in *T. neapolitana* subsp. *capnolactica* encouraged us to perform a bioinformatic analysis of the lactate dehydrogenase in the parent strain *T. neapolitana* (TnLDH) and among some taxonomically related members of the order Thermotogales. By using annotated LDH protein sequences, we identified putative genes in the open access genomes of *Thermotoga neapolitana* (*DSM4359* and *LA10*), *Thermotoga maritima*, *Thermotoga RQ7*, *Thermotoga naptophila* and *Pseudothermotoga lettingae*. In particular, one gene of LDH was identified from each species. A comparison of different amino acid L-LDH, D-LDH and L/MDH (Lactate/malate dehydrogenase) sequences from Thermotogales, LAB and other bacteria was performed using a phylogenetic approach. The resulting un-rooted phylogenetic tree led to a partition between two main branches according to L-LDH and D-LDH families, with TnLDH that clearly clustered in the L-LDH branch (Figure 2). According to a BLASTp approach, the consensus LDH sequence of

Thermotogales was compared to LDH from LAB, showing a percentage of identity and positive higher than 40% and 60%, respectively. Interestingly, Thermotogales LDH alignment highlighted the presence of an additional peptide MPSPCLYSITTEVIS in *T. neapolitana* (Figure S3). Using the TMHMM prediction server (http://www.cbs.dtu.dk/services/TMHMM) [15], we assigned a 60% probability of transmembrane localization to TnLDH for the presence of this peptide. No transmembrane domains were identified in the other LDH enzymes from Thermotogales.

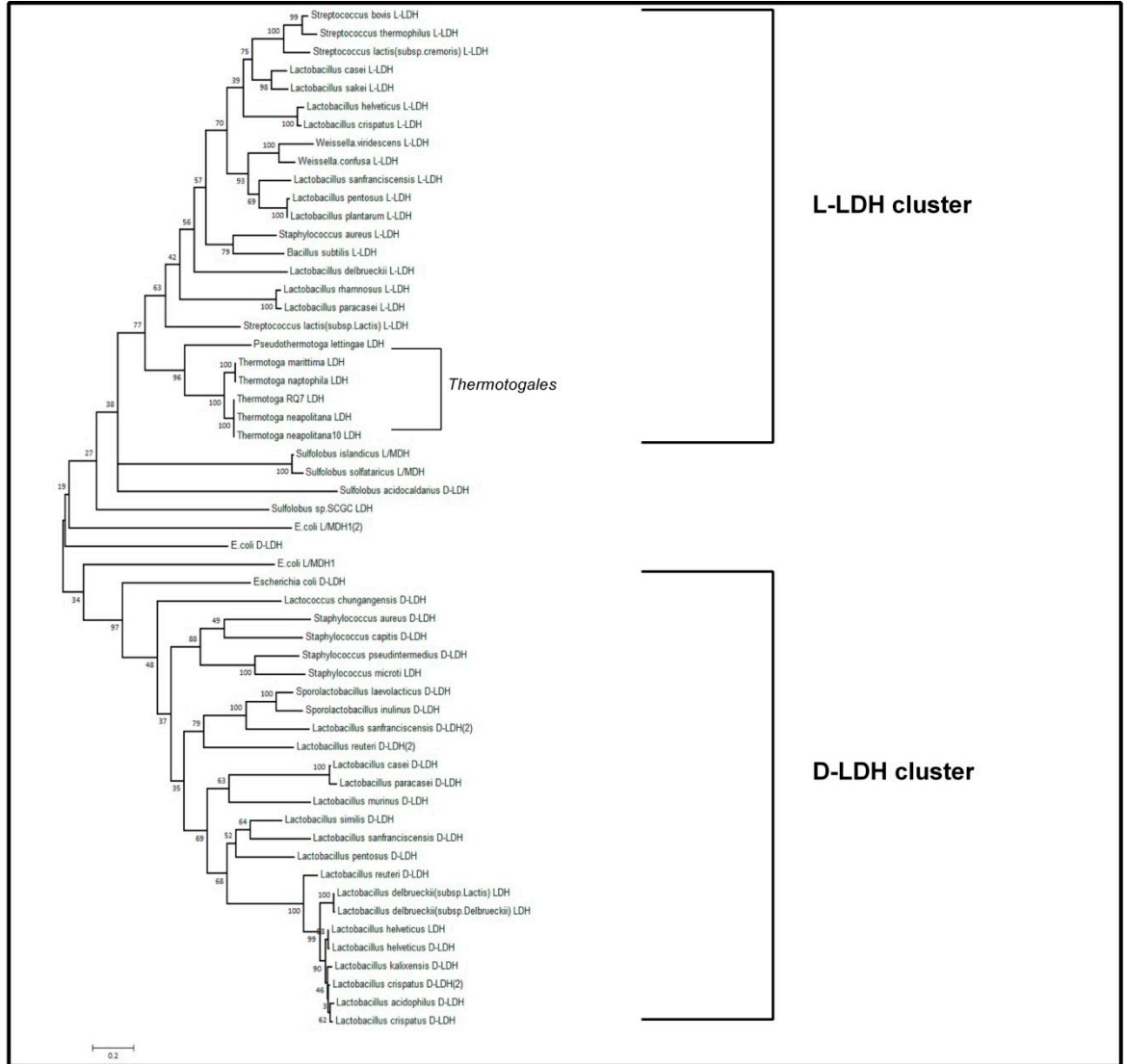

**Figure 2.** Un-rooted phylogenetic tree of lactate dehydrogenase amino acid sequences constructed using maximum likelihood method. The bootstrapping test (replicate = 100) is indicated on each node, in order to test the phylogeny.

## 4. Discussion

Lactic acid as a simple chiral molecule is widely used as a bulk chemical for the production of the green polymer PLA and other bioplastics. Polylactic acid is one of the most versatile biodegradable materials, which can be tailored into different resin grades for processing into a wide array of products. Depending on the composition of the optically active L- and D- enantiomers, PLA can crystallize into three forms (α, β, and γ) [16]. Thus, blending of the D- and L- enantiomers provides an effective method for controlling both the physical properties and the rate of biodegradation of PLA [17]. Pure poly-L-lactic acid (PLLA) has gained great attention because of its excellent biocompatibility and

mechanical properties. L-Polylactide is a semi-crystalline polymer exhibiting high tensile strength and low elongation with a high modulus that has recently been found to have new applications in medicine. According to the versatility of utilization, global demand for L-LA has been steadily increasing in the last years. Production of L-LA, and the D isomer, is primarily based on biotechnological fermentation of sugars. For this reason, there is a continuous search for new microorganisms to use in lactic fermentation, including thermostable strains [16,18].

In this paper we report for the first time that the CLF pathway of *T. neapolitana* is able to produce L-lactic acid with an e.e. above 95%. The stereochemistry assignment is confirmed by the bioinformatics analysis that allocates TnLDH and other related sequences of Thermotogales within the L-LDH sub-family. The classical GXGAGVG NADH/NAD$^+$ binding domain of the LDH is present in all these sequences as GL/AGRVG (22aa-27aa in *T. neapolitana*; 7aa-12aa in the others) [19]. However, sequence analysis of the protein of *T. neapolitana* shows the unconventional presence of a putative transmembrane peptide that may be related to a subcellular localization of this protein. This peptide is absent in the sequences of other Thermotogales. Lactate dehydrogenase of Thermotogales shows critical arginine and histidine in position R101, R174, R176 and H181, respectively [20]. The last residue plays a central role in fructose 1,6 bisphosphate binding and enzyme activity [21]. Analogously, the residues N133 and R174 in the protein of *T. neapolitana* are critical in the pyruvate binding [22].

In conclusion, the present work contributed to identifying the stereochemistry of lactic acid produced by *T. neapolitana* and to expanding the knowledge about potential in lactic acid production in thermophilic bacteria that offers considerable advantages for biotechnological processes. Furthermore, the current results highlight the effective significance of CLF as a suitable biotechnological pathway to give a high yield of optically pure L-lactate from a potential $CO_2$-based process. Further study will be necessary to elucidate the molecular peculiarity and the bioenergetics of this process.

**Supplementary Materials:** The following are available online at http://www.mdpi.com/2311-5637/5/2/34/s1, Figure S1: GC/MS of DL-Lactate, L-lactate, lactate of *T. neapolitana* and co-elution DL-Lactate and lactate of *T. neapolitana* as O-trifluoracetyl-(–)-menthyl ester, Figure S2: GC/MS lactate of *T. neapolitana* and integration of peak area of L- and D-lactate. Figure S3: LDH alignments of Thermotogales sequences.

**Author Contributions:** G.N.: stereochemical analysis and helped analyze the chemical results; S.L., bioinformatics; N.E., microbiology; E.M., stereochemical analysis; A.F. conceived the original idea and wrote the manuscript with support from G.N. and S.L.; G.d. supervised the project together with A.F.

**Funding:** This research was funded by BioRECO2VER_H2020NMBP-BIO-2017 project, grant number 760431.

**Acknowledgments:** The authors would like to thank Lucio Caso and Angela Sardo (CNR-ICB) for the technical support in microbiological activities.

**Conflicts of Interest:** The authors declare no conflict of interest.

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
