# Peer review of "Capnophilic Lactic Fermentation from Thermotoga neapolitana: A Resourceful Pathway to Obtain Almost Enantiopure L-lactic Acid"

_fermentation, doi:10.3390/fermentation5020034_

Reviewer 1 Report

Dear Authors,

Your manuscript needs intensive improvement. All my suggestions and errors that are found are signed in your manuscript.

Author Response

We sincerely appreciated the reviewer’s suggestions. We have made every change she/he required.

Detailed answers to the major issues are reported below:

REVIEW: The title should be re-edited for this form of a manuscript or necessary data about fermentation process should be added.

ANSWER: To improve the manuscript, additional data about fermentation process, bacterial growth and organic acid production were added in the paper.

REVIEW: I suggest to use the names of genera: Lactococcus spp. and Lactobacillus spp.

ANSWER: The names of genera (Lactococcus spp. and Lactobacillus spp.) were used to indicate LAB, as suggested.

REVIEW: It would be better present potential of L-enantiomer production and possibilities of using it. ANSWER: we agree with the reviewer. We are asking to include this work in a special issue about lactic acid and, we are sure that other papers will be committed on this point. On the other hand, we would like to keep the original focus on identification of the lactic acid in T. neapolitana and the potential of CLF as new pathway to exploit for this aim.

REVIEW: rather knowledge about the forms of lactic acid produced by T. neapolitana.

ANSWER: We have slightly modified the sentence as suggested from the review.

All minor mistakes (e.g. font for protein, abbreviation) were revised according to the review’s comments. English usage has also been refined.

Reviewer 2 Report

It was a pleasure to read such brief but sufficiently enough manuscript describing the novel and fresh science with properly described experiments and short but essential discussion part.

I've met some editorial mistakes (line 81, line 166) so You should carefully check the manuscript for more such mistakes - if are.  
In my opinion "introduction" part from Discussion (lines 155-159) should be removed since there are almost the same as in the Introduction

Author Response

We thank the reviewer for the nice comment. We read carefully the manuscript as suggested to eliminate typos and we have slightly modified the first part of the Discussion (lines 155-159) to avoid repetitions with the Introduction.

Round  2

Reviewer 1 Report

Dear Authors,

Your corrections have improved this manusctript significantly. However, some issues should be corrected.

My suggestions are in the text.

Author Response

Dear Dr. Kaitlyn Wu

we thank the reviewer for the time dedicated to our manuscript. Every suggested revision has been done. In Table 1, we added the information on biomass production in g/L (dry biomass) as requested by the reviewer.